# MINIMUM EDIT DISTANCE TRAINING FOR CONDITIONAL LANGUAGE GENERATION MODELS

## ABSTRACT

The utilization of attention-based encoder-decoder (AED) structures, including transformers, has further advanced the capabilities of conditional language generation (CLG) models. However, the conventional AED model training approach which aims to maximize the likelihood conditioned on the prefix of reference label sequence, introduces exposure bias and possesses limitations in that it uses different evaluation metrics in the training and inference stages. In this study, we introduce a novel AED model training technique focused on minimizing the Levenshtein distance between the reference and inferred label sequences. The proposed method effectively mitigates exposure bias and improves the generalization performance of neural machine translation and automatic speech recognition models. Furthermore, we demonstrate that a post-hoc calibration function trained with the proposed objective function significantly reduces the calibration error of the ASR model, resulting in notable performance improvements.

## 1 INTRODUCTION

Conditional language generation (CLG) tasks aim to find the optimal sequence of tokens $\hat{Y}$ for a given source context $X$.

$$\hat{Y} = \underset{Y}{\operatorname{\textbf{argmax}}} P(Y|X) \tag{1}$$

Depending on the composition of the source context and target token sequence, various tasks such as automatic speech recognition (ASR), neural machine translation (NMT), image captioning, and generative question answering are classified into CLG categories Cho et al. (2014); Bahdanau et al. (2015); Sutskever et al. (2014); Chorowski et al. (2015); Vinyals et al. (2015); Nenkova et al. (2011). In order to perform the CLG task well, it is important to approximate $P(Y|X)$ well. Today, neural network models trained on large paired datasets have exhibited notable performance. Among these models, recurrent neural network (RNN)-based sequence-to-sequence models and transformers, which are specialized in time-series data processing, have demonstrated exceptional capabilities in CLG tasks Cho et al. (2014); Bahdanau et al. (2015); Sutskever et al. (2014); Vaswani et al. (2017). Since direct modeling of $P(Y|X)$ is intractable, the following factorization process is required:

$$P(Y|X) = \prod_{t=1}^{T} P(Y_t|Y_{0:t-1}, X) \tag{2}$$

where $Y = [Y_1, ..., Y_T] = Y_{1:T}$, $Y_0$ denotes the *start of sentence token* and $T$ denotes the length of the token sequence. Most neural network-based CLG models update the parameter $\theta$ to minimize the negative-log-likelihood (NLL) of the factorized linear chain.

$$\mathsf{NLL}(X^i, Y^i; \theta) = -\sum_{t=1}^{T^i} \log P(Y_t^i|Y_{0:t-1}^i, X^i; \theta) \tag{3}$$

where $(X^i, Y^i)$ denotes $i$th paired sample in $n$-sized paired dataset $\{(X^i, Y^i)\}_{i=1}^{n}$ and $T^i$ denotes the length of $Y^i$. In the above equation, the CLG model is trained to maximize the likelihood computed conditionally on true previous tokens ($Y_{0:t-1}^i$). This training method is known as teacher forcing or simply maximum likelihood estimation (MLE), and the loss function in Equation 3 is referred to as next-word prediction loss. The teacher forcing method parallelizes the training process

of the autoregressive models, thereby increasing the efficiency of training and freeing the model from the effects of accumulated errors resulting from incorrect inferences of the previous time step during the training process Venkatraman et al. (2015). However, in the inference stage of CLG models, the previously inferred tokens $\hat{Y}_{0:t-1}^i$ are used instead of the true previous tokens to infer the token of the current time step. At this point, exposure bias occurs because of the difference between $P(Y)$ and $P(\hat{Y})$ Bengio et al. (2015); Ranzato et al. (2016). The exposure bias of the CLG model exacerbates hallucinations Wang & Sennrich (2020), and causes calibration errors, and error accumulation problems Wang et al. (2020); Arora et al. (2022). In addition, exposure bias is attracting attention as a cause of chronic problems in CLG models, such as lack of vocabulary diversity and word-level repetition Li et al. (2016); Welleck et al. (2019); Choi et al. (2020).

Another problem with the CLG model trained by teacher forcing is that the model evaluation criteria differ during the training and inference processes. While most CLG models utilize measures that evaluate decoding results at the sequence level, such as the *bilingual evaluation understudy* (BLEU) or *recall-oriented understudy for gisting evaluation* (ROUGE) score and *word error rate* (WER), the MLE in Equation 3 uses measures that evaluate candidates at the token level Papineni et al. (2002); LIN (2004); Wiseman & Rush (2016). We present the differences between the evaluation metric in the training and inference processes in more detail in Section 2.3. These two problems in CLG models are well-known, and numerous studies have been conducted to address them. Sampling-based methods that take some prefixes from $P(\hat{Y})$ in the training process and label smoothing methods that prevent giving an excessively high probability mass to reference label sequences are two simple heuristics used to alleviate these problems Müller et al. (2019); Wang et al. (2020). In addition, there are methods for training models by using sequence-level loss or reinforcement learning-based methods that directly minimize evaluation metrics Edunov et al. (2018); Wang & Sennrich (2020); Veselý et al. (2013); Zhao et al. (2022); Paulus et al. (2018). Our study shares the same motivation as existing studies in that it trains the CLG model in a way that overcomes exposure bias and directly minimizes evaluation measures. The contributions of this study can be summarized as follows:

1. We defined a Levenshtein distance-based alignment that is uniquely determined between two sequences of different lengths.

2. The alignment defined in Step 1 was used to design a new sequence-level loss function.

3. We train the NMT and ASR models using the proposed loss function and obtain the following results:

    (a) The proposed method outperforms other sequential model training strategies such as teacher forcing, scheduled sampling, and minimum Bayes risk (MBR) in ASR and NMT tasks.

    (b) The proposed method uses the autoregressively obtained decoding result as a prefix to train the CLG model and mitigate exposure bias. We experimentally demonstrated that the proposed method alleviates the accumulated errors caused by exposure bias.

    (c) We experimentally demonstrate that the post-hoc calibration function trained with the proposed loss function achieves a lower calibration error and higher performance gain.

## 2 BACKGROUND

### 2.1 EDIT DISTANCE

Edit distances serve as distance measures at the sequence level for assessing the similarity between pairs of sequences. The edit distance defines the number of operations needed to transform a source sequence into a target sequence as the distance between the two sequences. Edit operations encompass substitution, insertion, deletion, and transposition; specific types of edit distances permit only certain operations. A notable instance of edit distance is the Hamming distance. Hamming distance Hamming$(\cdot, \cdot)$ is defined as the number of substitutions required to convert a source sequence to a target sequence with the same lengths.

**Definition 2.1.** (Hamming distance) For two label sequences $Y_{1:T}$ and $\hat{Y}_{1:T}$ of length $T$, the Hamming distance is defined as follows:

$$\mathsf{Hamming}(Y_{1:T}, \hat{Y}_{1:T}) = \sum_{t=1}^{T} \mathbf{1}(Y_t \neq \hat{Y}_t) \tag{4}$$

where $\mathbf{1}$ denotes the indicator function. It is also possible to define the distance between a pair of sequences of different lengths using sequence-level edit operations such as insertion or deletion. The Levenshtein distance $\mathsf{Lev}(\cdot, \cdot)$ is a distance measure that utilizes three types of edit operations: insertion, deletion, and substitution.

**Definition 2.2.** (Levenshtein distance) Given a token sequence $Y_{1:T}$ of length $T$ and a token sequence $\hat{Y}_{1:\hat{T}}$ of length $\hat{T}$, the Levenshtein distance between the prefixes of the two sequences $Y_{1:t}$ and $\hat{Y}_{1:\hat{t}}$ is defined as follows:

$$\mathsf{Lev}(Y_{1:t}, \hat{Y}_{1:\hat{t}}) = \begin{cases} \mathsf{Lev}(Y_{1:t-1}, \hat{Y}_{1:\hat{t}-1}) & \text{if } Y_t = \hat{Y}_{\hat{t}} \\ 1 + \min \begin{cases} \mathsf{Lev}(Y_{1:t-1}, \hat{Y}_{1:\hat{t}-1}) \\ \mathsf{Lev}(Y_{1:t}, \hat{Y}_{1:\hat{t}-1}) \\ \mathsf{Lev}(Y_{1:t-1}, \hat{Y}_{1:\hat{t}}) \end{cases} & \text{otherwise} \end{cases} \tag{5}$$

where $t \in \{1, ..., T\}, \hat{t} \in \{1, ..., \hat{T}\}, \mathsf{Lev}(Y_0, \hat{Y}_{1:\hat{t}}) = \hat{t}, \mathsf{Lev}(Y_{1:t}, \hat{Y}_0) = t$ and $\mathsf{Lev}(Y_0, \hat{Y}_0) = 0$.

## 2.2 Evaluation Measures for CLG models

This study carried out experiments utilizing ASR and NMT models. NMT is the task of generating target language sentences corresponding to source language sentences, while ASR is the task of extracting language information inherent in given speech signals. Despite their commonality in dealing with time-series data, these tasks employ different evaluation measures. In NMT, $\mathcal{N}$-gram matching-based evaluation measures are predominantly used. A representative evaluation metric for NMT models is the BLEU score, which gauges sequence-level similarity based on the count of consecutive token sequences of length $\mathcal{N}$ shared between target and candidate token sequences Papineni et al. (2002).

A characteristic of the ASR, which differs from other CLG tasks, is that there are few reference label sequences corresponding to one speech feature. Except for homonyms and ambiguous transcriptions, most speech features have a unique token sequence as a target. Therefore, Levenshtein distance was used as an evaluation measure in the ASR task. Depending on the unit criterion to be evaluated, the word error rate or token error rate (TER) was used.

**Definition 2.3.** (Token error rate) Given a reference token sequence $Y_{1:T}$ and a candidate token sequence $\hat{Y}_{1:\hat{T}}$, the TER is defined as follows:

$$\mathrm{TER}(Y_{1:T}, \hat{Y}_{1:\hat{T}}) = \frac{\mathsf{Lev}(Y_{1:T}, \hat{Y}_{1:\hat{T}})}{T} \tag{6}$$

## 2.3 Discrepancy in Evaluation Criteria between Training and Inference

Given a source context $X$, well-trained CLG models generate a candidate label sequence $\hat{Y}$ resembling the reference label sequence $Y$. The sequence-level error of the CLG model trained in the teacher-forcing scenario can be quantified with Hamming distance as follows:

$$\mathrm{Error}_{\mathrm{train}}(X, Y_{1:T}; \theta) = \mathsf{Hamming}(Y_{0:T}, \hat{Y}_{1:T}) \tag{7}$$

where $\hat{Y}_t = \underset{y \in \mathcal{V}}{\operatorname{argmax}} P(y|X, Y_{1:t-1}; \theta), t \in \{1, ..., T\}, \theta$ denotes the parameters of the CLG model, and $\mathcal{V}$ denotes vocabulary. It's evident that the teacher-forcing-based MLE loss in Equation 3 compels CLG models to minimize the above error. The accuracy estimated during the training phase of the CLG model using MLE can be expressed as follows:

$$\mathrm{ACC}_{\mathrm{train}} = \frac{\#\mathsf{correct}}{T} = \frac{\#\mathsf{correct}}{\#\mathsf{correct} + \#\mathsf{substitution}} \tag{8}$$

While error and accuracy are defined as above in the training phase, CLG models typically employ sequence-level evaluation measures during the evaluation phase. The error in CLG models evaluated using the TER can be represented as:

$$\text{Error}_{\text{test}}(X, Y_{1:T}; \theta) = \text{Lev}(Y_{1:T}, \hat{Y}_{1:\hat{T}}) \tag{9}$$

where $\hat{Y}_{\hat{t}} = \underset{y \in \mathcal{V}}{\operatorname{argmax}} \, P(y|X, \hat{Y}_{0:\hat{t}-1}; \theta)$ and $\hat{t} \in \{1, ..., \hat{T}\}$. Accuracy estimated in the evaluation phase was also redefined as follows.

$$\text{ACC}_{\text{test}} = \frac{\#\text{correct}}{\#\text{correct} + \#\text{substitution} + \#\text{deletion} + \#\text{insertion}} \tag{10}$$

We hypothesized that this discrepancy between the training and evaluation phases deteriorates the generalization performance of CLG models. Based on this assumption, we propose a new objective function to train the CLG model to directly minimize $\text{Error}_{\text{test}}$.

## 3 PROPOSED METHODS

### 3.1 LEVENSHTEIN BACKWARD PATH (LBP)

The sequence-level error ($\text{Error}_{\text{test}}$) presented in Equation 9 is calculated between sequences of differing lengths. To design a loss function directly minimizing it, identifying a suitable alignment between the sequence pair is essential. In this regard, we define a path derived through the Levenshtein distance between sequence suffixes.

**Definition 3.1.** (LBP) LBP between a pair of sequences $\psi_{0,0}(Y_{1:T}, \hat{Y}_{1:\hat{T}})$ is defined as follows:

$$\psi_{t,\hat{t}} = \begin{cases} \psi_{t+1,\hat{t}+1} \cup (t+1, \hat{t}+1) & \text{if } Y_t = \hat{Y}_{\hat{t}} \\ \begin{cases} \psi_{t+1,\hat{t}+1} \cup (t+1, \hat{t}+1) & \text{if } \text{Lev}_{\min} = \frac{\text{Lev}(Y_{t+1:T}, \hat{Y}_{\hat{t}+1:\hat{T}})}{t+\hat{t}+2} \\ \psi_{t,\hat{t}+1} \cup (t, \hat{t}+1) & \text{if } \text{Lev}_{\min} = \frac{\text{Lev}(Y_{t:T}, \hat{Y}_{\hat{t}+1:\hat{T}})}{t+\hat{t}+1} \\ \psi_{t+1,\hat{t}} \cup (t+1, \hat{t}) & \text{if } \text{Lev}_{\min} = \frac{\text{Lev}(Y_{t+1:T}, \hat{Y}_{\hat{t}:\hat{T}})}{t+\hat{t}+1} \end{cases} & \text{otherwise} \end{cases}$$

where $\text{Lev}_{\min} = \min(\frac{\text{Lev}(Y_{t+1:T}, \hat{Y}_{\hat{t}+1:\hat{T}})}{t+\hat{t}+2}, \frac{\text{Lev}(Y_{t:T}, \hat{Y}_{\hat{t}+1:\hat{T}})}{t+\hat{t}+1}, \frac{\text{Lev}(Y_{t+1:T}, \hat{Y}_{\hat{t}:\hat{T}})}{t+\hat{t}+1})$, $t \in \{0, ..., T-1\}$, $\hat{t} \in \{0, ..., \hat{T}-1\}$, $\psi_{T,\hat{t}} = \{(T, \hat{t}), (T, \hat{t}+1), ..., (T, \hat{T})\}$, $\psi_{t,\hat{T}} = \{(t, \hat{T}), (t+1, \hat{T}), ..., (T, \hat{T})\}$, $\psi_{T,\hat{T}} = \varnothing$. In Figure 1, we depicted the edit distance matrix pertaining to the string pair ('DIVERS' and 'DRIVE') as a heatmap and highlighted LBP on it. This LBP encompasses four types of edit operations correct, insertion, deletion, and substitution. In Figure 1, we highlighted insertion errors on LBP in yellow, deletion errors in red, and correct in green.

### 3.2 MINIMUM EDIT DISTANCE TRAINING OVER THE LBP

We introduce a new training technique for CLG models minimizing the Levenshtein distance between a reference label sequence and a candidate sequence obtained via autoregressive decoding. The proposed method trains CLG models to maximize the likelihood computed over the LBP.

**Definition 3.2.** (MED) For the paired sample $(X^i, Y^i_{1:T^i}) \sim P(X, Y)$ and its beam search decoding result $\hat{\mathcal{Y}}^i_{1:\hat{T}^i}$, objective function $\text{NLL}_{\text{MED}}(X^i, Y^i_{1:T^i}, \hat{\mathcal{Y}}^i_{1:\hat{T}^i}; \theta)$ is defined as follows:

$$\text{NLL}_{\text{MED}}(X^i, Y^i, \hat{\mathcal{Y}}^i; \theta) = - \sum_{(t,\hat{t}) \in \psi^i} \log P(Y^i_t | \hat{\mathcal{Y}}^i_{0:\hat{t}-1}, X^i; \theta) \tag{11}$$

where $\psi^i = \psi^i_{0,0}(Y^i_{1:T^i}, \hat{Y}^i_{1:\hat{T}^i})$ and $\hat{Y}^i_{\hat{t}} = \underset{y \in \mathcal{V}}{\operatorname{argmax}} \, P(y|\hat{\mathcal{Y}}^i_{0:\hat{t}-1}, X^i; \theta)$. As shown in Figure 1, the MED loss trains the model to correct the three types of errors (insertion, deletion, and substitution) presented on the LBP. This results in the proposed method minimizing the Levensheitn distance-based error ($\text{Error}_{\text{test}}$) between the candidate and the reference label sequence.

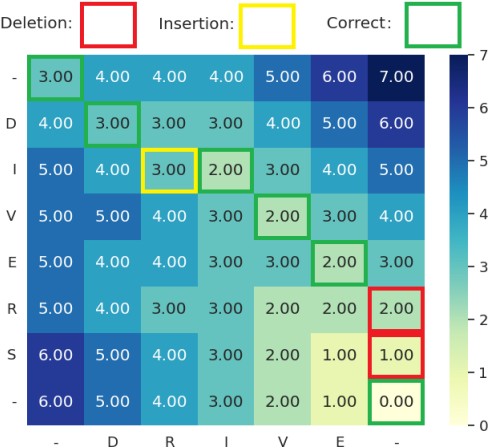

Figure 1: Edit distance matrix with Levenshtein backward path (LBP). The LBP in the matrix is highlighted by colored boxes.

The proposed training technique employs the beam search decoding results $\hat{\mathcal{Y}}^i$ as a prefix for likelihood calculation. $\hat{\mathcal{Y}}^i$ is pre-generated via an external model that has been trained already (offline scenario) or generated in real-time during each iteration using a model that is being trained (online scenario). The two training scenarios are illustrated in Figure 2. Among the beam-search decoding results, we chose the sentence with the highest score; however, if the TER of the sentence exceeded the threshold $\xi_{\mathrm{TER}}$, it was excluded from MED training. We obtained higher performance when $\xi_{\mathrm{TER}}$ was scheduled therefore, we set $\xi_{\mathrm{TER}} = \xi_{\max} \frac{\lambda}{\lambda_{\max}}$ in all experiments. $\lambda$ denotes the current training epoch, and $\lambda_{\max}$ denotes the last training epoch. Unless otherwise specified, we used 1.0 as the $\xi_{\max}$ value. Through this scheduling method, it is possible to prevent the phenomenon in which the convergence speed is degraded owing to erroneous decoding results generated in the initial stage of training in an online scenario.

Because the inferred label sequence $\hat{Y}_{1:\hat{t}}^i$ is used to calculate the LBP, the CLG model trained using only $\mathrm{NLL_{MED}}$ in an offline scenario has a trivial solution that outputs the input label of the current time step. We avoid this problem by using a multi-task learning approach that minimizes NLL and $\mathrm{NLL_{MED}}$ together. Finally, we updated the CLG model parameters to minimize the objective function as follows:

$$\mathcal{L}_{\mathrm{MED}}(X^i, Y^i, \hat{\mathcal{Y}}^i; \theta) = \mathrm{NLL}(X^i, Y^i; \theta) + \mathrm{NLL_{MED}}(X^i, Y^i, \hat{\mathcal{Y}}^i; \theta) \tag{12}$$

### 3.3 SEGMENT-LEVEL LEVENSHTEIN BACKWARD PATH (SLBP)

In many CLG tasks, there can be multiple semantically similar token sequences corresponding to a single source context. In these tasks, proper alignment may not exist because of the morphological differences between $Y^i$ and $\hat{\mathcal{Y}}^i$. In this section, we propose a minimum edit distance training method for more relaxed conditions. Given a sentence $Y^i = Y_{0:T^i}^i$ and an integer $\mathcal{N} \in \{1, ..., T^i\}$, we define $\mathcal{N}$-gram segments of $Y^i$ as $[G(Y^i, \mathcal{N})] = [Y_{1:\mathcal{N}}^i, Y_{2:\mathcal{N}+1}^i, \ldots, Y_{T-\mathcal{N}+1:T}^i]$. We find the set of segment pairs $\Omega^i(Y^i, \hat{Y}^i; \mathcal{N}, \xi_{\mathcal{N}})$ with the smallest edit distance from each of all $\mathcal{N}$-gram segments of $Y^i$ to the $\mathcal{N}$-gram segments of $\hat{Y}^i$.

$$\begin{aligned}
\Omega^i &= \{\cup_{u=1}^{T^i+1-\mathcal{N}} (u, \hat{u}) | \mathrm{Lev}([G(Y^i, \mathcal{N})]_u, [G(\hat{Y}^i, \mathcal{N})]_{\hat{u}}) \\
&= \min(\{\xi_{\mathcal{N}}\} \cup \{\mathrm{Lev}([G(Y^i, \mathcal{N})]_u, [G(\hat{Y}, \mathcal{N})]_l)\}_{l=1}^{\hat{T}^i+1-\mathcal{N}})\},
\end{aligned} \tag{13}$$

where $\xi_{\mathcal{N}} \geq 0$ denotes the minimum edit distance threshold and $[G(Y^i, \mathcal{N})]_u = Y_{u:u-1+\mathcal{N}}^i$. We set $\mathcal{N}$ as the value IID sampled from the discrete uniform distribution $\mathcal{U}(2, 6)$ for every iteration and set $\xi_{\mathcal{N}}$ as the quotient of $\mathcal{N}$ divided by 2. We create a new tuple set (sLBP) by finding the LBP for each pair of segments in $\Omega^i$.

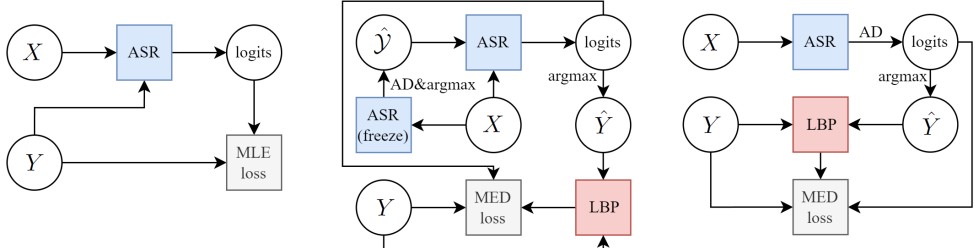

Figure 2: Proposed MED training scenarios. Left: Teacher forcing-based MLE. Middle: Offline MED training scenario. Right: Online MED training scenario. "AD" denotes autoregressive decoding (e.g., beam search decoding).

**Definition 3.3.** (sLBP) sLBP $\phi^i(Y^i, \hat{Y}^i; \mathcal{N}, \xi_{\mathcal{N}})$ is defined as follows:

$$\phi^i = \cup_{(u,\hat{u}) \in \Omega^i(Y^i, \hat{Y}^i; \mathcal{N}, \xi_{\mathcal{N}})} \cup_{(t,\hat{t}) \in \psi_{0,0}^i([G(Y^i, \mathcal{N})]_u, [G(\hat{Y}^i, \mathcal{N})]_{\hat{u}})} (u + t, \hat{u} + \hat{t}) \qquad (14)$$

### 3.4 Minimum Edit Distance Training over the sLBP

We present segment-level minimum edit distance training (sMED), a technique for training CLG models to maximize the likelihood on $\cup_{i=1}^n \phi^i$. Similar to the evaluation measures based on $\mathcal{N}$-gram matching, sMED assumes that the properly inferred token sequence $\hat{Y}$ corresponding to the source context shares many $\mathcal{N}$-gram segments with the reference label sequence $Y$. Therefore, the proposed method trains the CLG model to minimize the Levenshtein distance between similar $\mathcal{N}$-gram segments shared by $Y$ and $\hat{Y}$.

**Definition 3.4.** (sMED) For the paired sample $(X^i, Y_{1:T^i}^i) \sim P(X, Y)$ and its beam search decoding result $\hat{\mathcal{Y}}_{1:\hat{T}^i}^i$, the objective function $\mathrm{NLL}_{\mathrm{sMED}}(X^i, Y_{1:T^i}^i, \hat{\mathcal{Y}}_{1:\hat{T}^i}^i; \theta, \mathcal{N}, \xi_{\mathcal{N}})$ is defined as follows:

$$\mathrm{NLL}_{\mathrm{sMED}}(X^i, Y^i, \hat{\mathcal{Y}}^i; \theta, \mathcal{N}, \xi_{\mathcal{N}}) = - \sum_{(t,\hat{t}) \in \phi^i} \log P(Y_t^i | \hat{\mathcal{Y}}_{0:\hat{t}-1}^i, X^i; \theta) \qquad (15)$$

where $\phi^i = \phi^i(Y^i, \hat{Y}^i; \mathcal{N}, \xi_{\mathcal{N}})$ and $\hat{Y}_{\hat{t}}^i = \underset{y \in \mathcal{V}}{\mathrm{argmax}}\, P(y | \hat{\mathcal{Y}}_{0:\hat{t}-1}^i, X^i; \theta)$. We trained the CLG model to minimize both NLL and $\mathrm{NLL}_{\mathrm{sMED}}$ using a multi-task learning approach.

$$\mathcal{L}_{\mathrm{sMED}}(X^i, Y^i, \hat{\mathcal{Y}}^i; \theta, \mathcal{N}, \xi_{\mathcal{N}}) = \mathrm{NLL}(X^i, Y^i; \theta) + \mathrm{NLL}_{\mathrm{sMED}}(X^i, Y^i, \hat{\mathcal{Y}}^i; \theta, \mathcal{N}, \xi_{\mathcal{N}}) \qquad (16)$$

Because $\mathrm{NLL}_{\mathrm{MED}}$ and $\mathrm{NLL}_{\mathrm{sMED}}$ use the beam search decoding candidate $\hat{\mathcal{Y}}$ for model training, it is affected by the accuracies of these candidates. We did not use candidates with errors above the $\xi_{\mathrm{TER}}$ for training. We calculated the loss using a smaller $\mathcal{N}$ value when the reference label sequence and the candidate did not have any shared segments at the $\mathcal{N}$-gram level. These filtering schemes are shown in Algorithm 1 of the Appendix.

### 3.5 Calibration Function Training with MED

If the lengths of the inferred and reference label sequences are different, calibration error measures such as the expected calibration error (ECE), cannot be estimated because the token-wise accuracy is not defined Guo et al. (2017). In Wang et al. (2020), the calibration error between two sequences of different lengths was quantified using an edit distance. Using the LBP defined in 4.1, the calibration error measure proposed in Wang et al. (2020) is formulated concisely.

**Definition 3.5** (ECE$_{\text{LBP}}$). For $n$ paired samples and their best beam search decoding results $D_{\text{seq}} = \{(X^i, Y^i, \hat{\mathcal{Y}}^i)\}_{i=1}^n$, ECE$_{\text{LBP}}$ with $B$ bins evaluated on $D_{\text{seq}}$ is defined as follows:

$$\text{ECE}_{\text{LBP}}(D_{\text{seq}}, B) = \sum_{b=1}^{B} \frac{|\Psi_b|}{|\Psi|} |\text{ACC}_{\text{LBP}}(\Psi_b) - \text{CONF}_{\text{LBP}}(\Psi_b)|,$$

$$\text{ACC}_{\text{LBP}}(\Psi_b) = |\Psi_b|^{-1} \sum_{(i,t,\hat{t}) \in \Psi_b} \mathbf{1}(Y_t^i = \hat{Y}_{\hat{t}}^i),$$

$$\text{CONF}_{\text{LBP}}(\Psi_b) = |\Psi_b|^{-1} \sum_{(i,t,\hat{t}) \in \Psi_b} P(\hat{Y}_{\hat{t}}^i | \hat{\mathcal{Y}}_{1:\hat{t}-1}^i, X^i; \theta),$$

$$\Psi_b = \{(i, t, \hat{t}) \in \Psi : P(\hat{Y}_{\hat{t}}^i | \hat{\mathcal{Y}}_{1:\hat{t}-1}^i, X^i; \theta) \in (\frac{(b-1)}{B}, \frac{b}{B}]\}$$

(17)

where $\Psi = \cup_{i=1}^n \cup_{(t,\hat{t}) \in \psi^i} (i, t, \hat{t})$, $\hat{Y}_{\hat{t}}^i$ is the estimated label sequence conditioned on $(\hat{\mathcal{Y}}_{0:\hat{t}-1}^i, X^i; \theta)$ and $|\Psi_b|$ denotes the size of bin. As shown in the aforementioned equation, the calibration error measure of the CLG model proposed in Wang et al. (2020) can be interpreted as the difference between the accuracy and confidence estimated on LBP (ACC$_{\text{LBP}}$ is equal to ACC$_{\text{test}}$ estimated per bin). Accordingly, a post-hoc calibration function trained to minimize the MED loss on the validation set directly minimizes ECE$_{\text{LBP}}$. In Section 5, we show that the post-hoc calibration function trained with MED loss produces a well-calibrated output compared to the calibration function trained with MLE Kumar & Sarawagi (2019) in the inference stage, and consequently improves the beam-search decoding result.

## 4 EXPERIMENTS

We used FairSeq in the NMT experiments and ESPNet in the ASR experiments Watanabe et al. (2018); Ott et al.. We used label smoothing (with a label smoothing factor of 0.1) for all NMT model training. We trained the NMT model with the online scenario shown in Figure 2, and the ASR model with the offline scenario. The performance of the NMT model trained using the scheduled sampling method is presented in Table 1. In these experiments, we sampled the prefix from an interpolated distribution $\epsilon_{\text{SS}} P(\hat{Y}) + (1 - \epsilon_{\text{SS}}) P(Y)$ created with an interpolation factor $\epsilon_{\text{SS}}$ Bengio et al. (2015). We linearly increased the interpolation factor for every epoch $\lambda$ up to the maximum epoch $\lambda_{\text{max}}$.

$$\epsilon_{\text{SS}} = \epsilon_{\text{max}} \frac{\lambda}{\lambda_{\text{max}}}$$

(18)

We conducted experiments on $\epsilon_{\text{max}} \in \{0.1, 0.25, 0.5, 0.9\}$, and the experiments with the highest performance are presented in Table 1. In all calibration experiments, we utilized the temperature-scaling method Guo et al. (2017). We determined the optimal temperature value by gradient descent using a validation set, and in this process, the CLG model parameters were frozen.

### 4.1 DATA

In the NMT experiment, the IWSLT14 (En-De), IWSLT17 (En-Fr), and the WMT14 (En-De) datasets were employed, which consist of English, French, and German Cettolo et al. (2014). The IWSLT14 dataset encompassed a trainset of 160k, a validation set of 7k, and a test set of 6k instances. We adopted the byte pair encoding (BPE)-based tokenization technique, resulting in a composition of 6.6k tokens for English and 8.8k tokens for German Sennrich et al. (2016). The IWSLT17 dataset's training set constituted 226k sentence pairs and it is tokenized with BPE as 7k subwords for English and 9k subwords for French. The WMT dataset's training set constituted 4.5 million sentence pairs, with newstest13 and newstest14 serving as the validation and test sets, respectively. The WMT dataset was tokenized with 40k BPE tokens for English and 43k BPE tokens for German.

In our ASR experiments, the LibriSpeech dataset was used Panayotov et al. (2015). The LibriSpeech training set consists of 960h and was divided into three partitions 100h, 360h, and 500h. We used only the 100h dataset for our experiments. The LibriSpeech evaluation set was divided into two

Table 1: Training results of the NMT and ASR models. Superscript 1 denotes the results in Wang & Sennrich (2020), and superscript 2 denotes the results in Vaswani et al. (2017).

| data | training method | BLEU4 | |
|---|---|---|---|
| | | avg | std |
| IWSLT14 (De→En) | MLE[1] | 34.7 | - |
| | MBR[1] | 35.2 | - |
| | MLE | 34.65 | 0.12 |
| | Scheduled sampling ($\epsilon_{max} = 0.10$) | 35.03 | 0.09 |
| | sMED | 35.73 | 0.07 |
| IWSLT14 (En→De) | MLE | 28.43 | 0.07 |
| | Scheduled sampling ($\epsilon_{max} = 0.10$) | 28.68 | 0.04 |
| | sMED | 29.51 | 0.06 |
| IWSLT17 (En→Fr) | MLE | 41.76 | 0.06 |
| | Scheduled sampling ($\epsilon_{max} = 0.25$) | 42.03 | 0.12 |
| | sMED | 42.40 | 0.08 |
| IWSLT17 (Fr→En) | MLE | 41.27 | 0.10 |
| | Scheduled sampling ($\epsilon_{max} = 0.50$) | 41.32 | 0.11 |
| | sMED | 41.46 | 0.10 |
| WMT14 (En→De) | MLE[2] | 28.4 | - |
| | MLE | 28.47 | 0.14 |
| | Scheduled sampling ($\epsilon_{max} = 0.25$) | 28.68 | 0.04 |
| | sMED | 28.86 | 0.06 |
| data | training method | WER (%) | |
| | | test_clean | test_other |
| LibriSpeech (100h) | MLE | 5.4 | 14.8 |
| | MED | 4.9 | 13.8 |

validation (dev_clean and dev_other) and two test sets (test_clean and test_other). In ASR experiments, 300 tokenized subwords were used as recognition units Kudo & Richardson (2018). The raw speech signal was converted into 80-dimensional f-bank features, and the sampling rate was reduced to 1/6 through two CNN layers. For the language model training, we used the LibriCorpus dataset, which consists of 40M sentences extracted from books. The LibriCorpus dataset was tokenized using the 300-subword vocabulary used in the ASR training. We trained models for 70 epochs on the IWSLT14 and IWSLT17 datasets and 100 epochs each on the WMT and LibriSpeech datasets.

## 4.2 MODELS

The ASR models consist of two 2d-CNN-based subsampling layers, a 12-layer transformer encoder, and a 6-layer transformer decoder and were trained in a multi-task learning scenario using AED loss and auxiliary connectionist temporal classification loss Kim et al. (2017). The NMT model used in our experiments with the WMT14 dataset has a transformer structure, and we used the same hyperparameter setup as the transformer (big) model in Vaswani et al. (2017). We have shown the model structure in more detail as a table in Appendix A.

## 4.3 EVALUATION

A beam width of 20 was employed to evaluate the NMT model, while five was used for the ASR model. We used a beam width of 3 for the NMT models and 20 for the ASR models to generate the beam-search candidate $\hat{\mathcal{Y}}$. In the decoding process of ASR models, we summed the scores of the recurrent neural network (RNN)-based external language model trained with Libri Corpus through the shallow fusion Mikolov et al. (2010). We used the weight average method in ASR experiments. We saved the model weights for each epoch and evaluated each model by calculating the token accuracy for a validation set. The weights from five models demonstrating the highest accuracy on the validation set were averaged and employed for evaluation. All NMT experiments were repeated

Table 2: WER and $\text{ECE}_{\text{LBP}}$ of the calibrated ASR models according to the calibration function training method.

| loss | test-clean | | test-other | |
|---|---|---|---|---|
| | WER (%) | $\text{ECE}_{\text{LBP}}$ | WER (%) | $\text{ECE}_{\text{LBP}}$ |
| - | 5.4 | 0.046 | 14.8 | 0.128 |
| MLE | 5.1 | 0.036 | 14.3 | 0.099 |
| MED | 5.0 | 0.018 | 14.1 | 0.055 |

Table 3: $\text{ACC}_{\text{test}}$ of the ASR models per position measured on the LibriSpeech test set (test_other). We divided it into four parts according to the length of the prefix and presented the average accuracy.

| | $\hat{t} < 50$ | $50 \leq \hat{t} < 100$ | $100 \leq \hat{t} < 150$ | $150 \leq \hat{t}$ |
|---|---|---|---|---|
| MLE | 82.13% | 82.87% | 80.24% | 57.53% |
| MED | 82.21% | 83.22% | 81.18% | 64.91% |
| relative accuracy gain (%) | 0.10% | 0.42% | 1.17% | 12.83% |

thrice, with the seed altered each time. The average and standard deviation of the computed BLEU4 scores were then presented.

## 5   RESULTS

The evaluation results for the ASR and NMT models trained using the proposed method are presented in Table 1. This approach demonstrated superior generalization performance compared to the scheduled sampling and MBR loss-based methods presented in Wang & Sennrich (2020), both of which utilized the same transformer-based baseline. Our method used a beam search candidate as a prefix and depends on the accuracy of the candidates. This dependency was controlled by adjusting hyperparameters ($\xi_{\text{TER}}$, $\mathcal{N}$, and beam width). We presented the generalization performance of models trained with more diverse hyperparameter settings in Appendix D.

We conducted experiments to calibrate the calibration error of the trained ASR model using a temperature-scaling method Guo et al. (2017). The results of utilizing MLE and MED are showcased in Table 2 to ascertain the optimal temperature. MED effectively halved $\text{ECE}_{\text{LBP}}$ compared to MLE and concurrently enhanced the ASR model's generalization performance. These findings indicated that the calibration function trained with the MED loss provided a better approximation of the actual probability distribution. We presented more details of the calibration function training methods, evaluation results about out-of-distribution (OOD) datasets and reliability diagrams in Appendix C.

In Arora et al. (2022); Lin et al. (2021), a method quantifying exposure bias through accumulated error was introduced. We measured the accumulated error for each position of the ASR model output using $\text{ACC}_{\text{test}}$ from Equation 10, with the results listed in Table 3. The ASR model exhibited performance degradation at the end of lengthy sentences. Notably, models trained with MED showcased higher accuracy than those trained with MLE, and this accuracy gap widened with increasing prefix length. These experimental outcomes suggest that the MED loss effectively mitigates accumulated error attributed to exposure bias.

## 6   CONCLUSION

We introduced the LBP, a deterministically derived path between two sequences. Moreover, we put forth a novel loss function designed to maximize the likelihood of CLG models estimated over the LBP. The MED loss, introduced in Section 3.2, effectively alleviates the exposure bias and maintains consistent evaluation criteria throughout both the training and evaluation phases. Comparative experiments in ASR and NMT tasks demonstrated that the MED loss yields enhanced generalization performance compared to MLE. Notably, the MED loss significantly reduces calibration error when compared to MLE in experiments involving the training of the post-hoc calibration function.

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

## A  REPRODUCIBILITY

### A.1  HARDWARE SETTING

We used four Nvidia A100 in parallel to train the NMT models on the WMT14 dataset. We used a single A100 GPU for NMT model training on the IWSLT17 and IWSLT14 datasets. ASR experiments were performed with a single RTX3090 GPU.

### A.2  SOURCE CODE

We uploaded our code for the LBP generation algorithm as supplementary material. We showed the hyperparameter settings of the transformer models we used in Table 5. All notations in Table 5 are the same as in Vaswani et al. (2017).

## B  RELATED WORKS

The proposed method shares similar motivations with sequence-level training or scheduled sampling techniques. It employs decoded candidates sampled from $P(\hat{Y}|X;\theta)$ to formulate a loss function and mitigate exposure bias Veselỳ et al. (2013); Liu et al. (2022). The proposed method differs from sequence-level training approaches by defining an edit distance-based error (insertion, deletion, and substitution) and training the model to rectify it Edunov et al. (2018); Zhao et al. (2022); Paulus et al.

(2018). Notably, while scheduled sampling trains the model to minimize the Hamming distance between the reference and inferred label sequences, the proposed method trains the model to minimize the Levenshtein distance between these sequences Bengio et al. (2015); Venkatraman et al. (2015).

The proposed method bears similarities to training techniques that directly minimize evaluation metrics through reinforcement learning Sabour et al. (2018); Gu et al. (2019); Paulus et al. (2018). In Sabour et al. (2018), a reinforcement learning-based CLG model training method was introduced aiming to minimize the Levenshtein distance between suffixes of the target and inferred sequences. The proposed method distinguishes itself by determining a single path with the minimum Levenshtein distance, optimizing the likelihood of this determined path, and defining the loss at a segment level so that it can be applied to CLG tasks with multiple target sequences.

The LBP defined in Section 3.1 aligns with the alignment search technique utilized to quantify calibration error with the autoregressive decoding results of the CLG model in Wang et al. (2020). Notably, our study demonstrates the direct minimization of $\mathrm{ECE_{LBP}}$ through MED training, whereas in Wang et al. (2020), the calibration error was alleviated by using the adaptive label smoothing.

## C  CALIBRATION OF CLG MODELS

Various methods have been studied to alleviate calibration errors. These methods can be classified into two categories. The first category is methods applied during the model training process (label smoothing or sequence-level training are typical examples) Müller et al. (2019); Zhao et al. (2022); Veselỳ et al. (2013); Paulus et al. (2018); Edunov et al. (2018). The second category is a method of calibrating the output probability distribution of a trained model through post-processing and is called post-hoc calibration methods. As post-hoc calibration methods, isotonic regression, and Platt scaling are general examples Zadrozny & Elkan (2002). The temperature scaling is an extension of the Platt scaling to multi-class classification Guo et al. (2017). The scaling factor (temperature, $\mathcal{T}$) is trained to maximize the likelihood on a validation set. In this process, the model parameters are frozen, and the output probability distribution of the model is calibrated through the following temperature scaling function:

$$\hat{P}(x_i) = [\sigma(\frac{z_i}{\mathcal{T}})] \tag{19}$$

where, $z_i \in \mathbb{R}^K$ denotes the logit vector for $x_i$, $K$ denotes the number of calsses and $\sigma$ denotes softmax function. As can be seen from the above equation, a temperature value greater than 1 increases the entropy of the calibrated output probability distribution $\hat{P}(\hat{x}_i)$ and, in the opposite case, lowers the entropy.

Guo et al. (2017) presented a way to find the optimal temperature $\hat{\mathcal{T}}$ for a scalar classification model by solving the following convex optimization problem:

$$\hat{\mathcal{T}} = \underset{\mathcal{T}}{\textbf{argmin}} \sum_{i=1}^{n} \sum_{k=1}^{K} -\mathbf{1}(y_i = k) \cdot \log([\sigma(\frac{z_i}{\mathcal{T}})]_k) \tag{20}$$

where, $\{(x_i, y_i)\}_{i=1}^{n}$ denotes validation set. In Kumar & Sarawagi (2019); Lee & Chang (2021), the teacher forcing method was used to calibrate the output probability distribution of the CLG models.

$$\hat{\mathcal{T}} = \underset{\mathcal{T}}{\textbf{argmin}} \sum_{i=1}^{n} \sum_{t=1}^{T^i} \sum_{k=1}^{K} -\mathbf{1}(Y_t^i = k) \cdot \log([\sigma(\frac{[Z_t^i]}{\mathcal{T}})]_k) \tag{21}$$

where, $[Z_t^i] = f(Y_{0:t-1}^i, X^i; \theta)$ denotes unnormalized CLG model output. The above equation finds a scaling factor that minimizes the cross entropy between the one-hot encoded true label distribution and the model's output probability distribution for the validation set, and as a result, the CLG model's confidence approximates $\mathrm{ACC_{train}}$. We insist that confidence should approximate accuracy considering sequence level errors ($\mathrm{ACC_{test}}$). To this end, we train the calibration function using the MED loss proposed in section 3.2.

$$\hat{\mathcal{T}} = \underset{\mathcal{T}}{\operatorname{argmin}} \sum_{i=1}^{n} \sum_{(t,\hat{t}) \in \psi^i} \sum_{k=1}^{K} -\mathbf{1}(Y_t^i = k) \log([\sigma(\frac{[\hat{Z}_{\hat{t}}^i]}{\mathcal{T}})]_k) \qquad (22)$$

where, $[\hat{Z}_{\hat{t}}^i] = f(\hat{\mathcal{Y}}_{0:\hat{t}-1}^i, X^i; \theta)$. We conducted an evaluation using the WSJ dataset to prove that the proposed calibration function training method has an effect on the OOD dataset. The proposed method improved calibration error and generalization performance even for OOD datasets (Table 4). We showed the result of calibrating the output probability distribution of the ASR models as a reliability diagram in Figure 3 (in distribution, LibriSpeech), Figure 4 (OOD, test_eval92), and Figure 5 (OOD, test_eval93).

## D    FILTERING SCHEME

As shown in Algorithm 1, we did not use beam-search decoding results with errors above the $\xi_{\mathrm{TER}}$ for training. We calculated the sMED loss using a smaller $\mathcal{N}$ value when the reference label sequence and the candidate did not have any shared segments at the $\mathcal{N}$-gram level. We showed the evaluation results of the NMT model according to the application of the filtering scheme in Table 6. When candidates with high TER are not removed at all ($\xi_{\mathrm{TER}} = \infty$), sMED loss showed performance degradation. However, with an appropriate filtering scheme, the proposed sMED loss exhibited robust training across diverse hyperparameter settings and consistently achieved BLEU scores surpassing the baseline (bottom two rows).

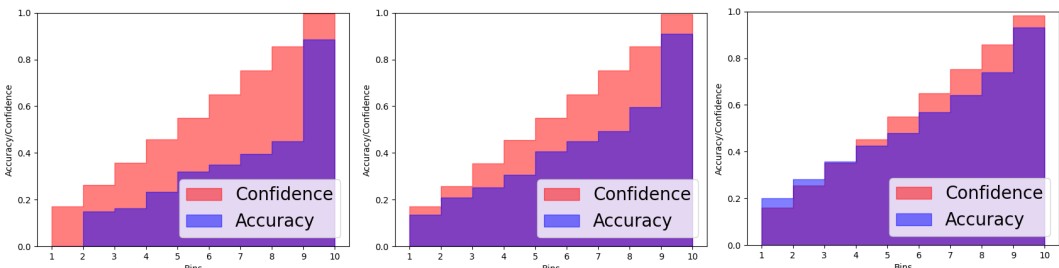

Figure 3: Reliability diagram estimated on LibriSpeech test_other (in distribution) with trained ASR model. Left: Uncalibrated, Middle: Calibrated with temperature scaling function trained with MLE, Right: Calibrated with temperature scaling function trained with MED.

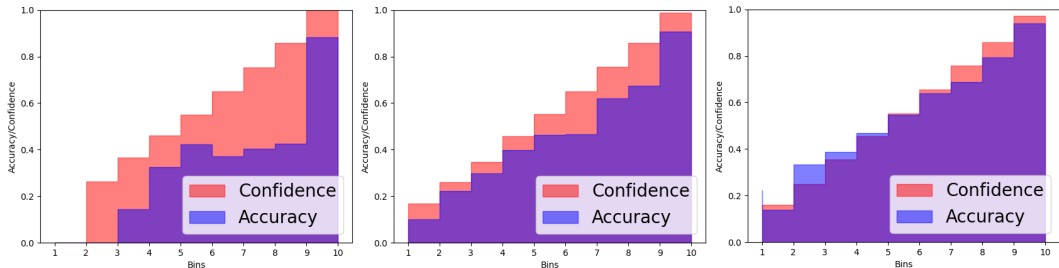

Figure 4: Reliability diagram estimated on WSJ test_eval92 (out of distribution) with trained ASR model. Left: Uncalibrated, Middle: Calibrated with temperature scaling function trained with MLE, Right: Calibrated with temperature scaling function trained with MED.

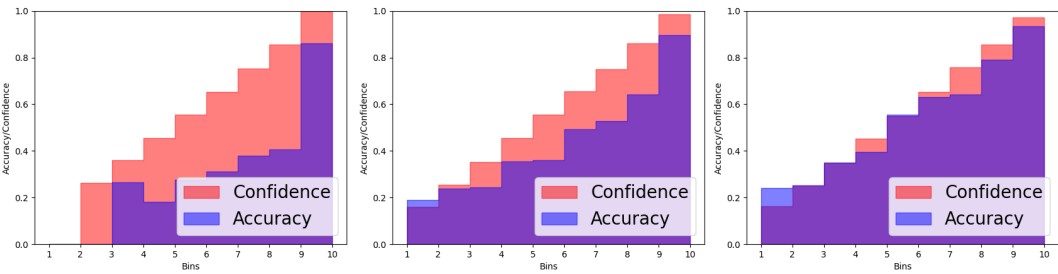

Figure 5: Reliability diagram estimated on WSJ test_eval93 (out of distribution) with trained ASR model. Left: Uncalibrated, Middle: Calibrated with temperature scaling function trained with MLE, Right: Calibrated with temperature scaling function trained with MED.

---

**Algorithm 1** Filtering Scheme for sMED Objective

---

**Input**: Paired data $(Y^i, X^i)$ and its decoding result $\hat{\mathcal{Y}}^i$
**Parameter**: $\theta, \mathcal{N}_{\min}, \mathcal{N}_{\max}, \lambda, \lambda_{\max}$
**Output**: sMED loss
1: $\mathcal{N} \sim \mathcal{U}(\mathcal{N}_{\min}, \mathcal{N}_{\max})$
2: $\xi_{\mathcal{N}} \leftarrow \frac{\mathcal{N}}{2}, \xi_{\mathrm{TER}} \leftarrow \frac{\lambda}{\lambda_{\max}}$
3: **if** $\mathrm{TER}(Y^i, \hat{\mathcal{Y}}^i) < \xi_{\mathrm{TER}}$ **then**
4:    $\hat{Y}^i \leftarrow \underset{Y}{\mathrm{argmax}}\, P(Y | \hat{\mathcal{Y}}^i, X^i; \theta)$
5:    **while** $\Omega^i(Y^i, \hat{Y}^i; \mathcal{N}, \xi_{\mathcal{N}}) = \varnothing$ and $\mathcal{N} > 1$ **do**
6:       $\mathcal{N} \leftarrow \mathcal{N} - 1$
7:    **end while**
8:    **return** $\mathrm{NLL}_{\mathrm{sMED}}(X^i, Y^i, \hat{\mathcal{Y}}^i; \theta, \mathcal{N}, \xi_{\mathcal{N}})$
9: **else**
10:    **return** 0
11: **end if**

---

Table 4: Calibration error and word error rate for out-of-distribution dataset .

|  | test_eval92 | | test_eval93 | |
| --- | --- | --- | --- | --- |
| loss | WER (%) | ECE | WER (%) | ECE |
| - | 14.7 | 0.122 | 19.2 | 0.147 |
| MLE | 14.1 | 0.086 | 18.5 | 0.103 |
| MED | 14.0 | 0.035 | 18.3 | 0.044 |

Table 5: Hyperparametter setting. 2d-CNN denotes the number of subsampling layers in ASR models, $d_{\mathrm{model}}$ denotes dimension of attention module, $d_{\mathrm{ff}}$ denotes dimension of feed forward networks, and $h$ denotes the number of attention heads.

| task | dataset | # 2d-CNN | Encoder layer | Decoder layer | $d_{\mathrm{model}}$ | $d_{\mathrm{ff}}$ | $h$ |
| --- | --- | --- | --- | --- | --- | --- | --- |
| NMT | IWSLT14,IWSLT17 | - | 6 | 6 | 512 | 1024 | 4 |
| | WMT14 | - | 6 | 6 | 1024 | 4096 | 16 |
| ASR | LibriSpeech | 2 | 12 | 6 | 256 | 2048 | 4 |

Table 6: Evaluation of NMT models on IWSLT14 (De2En) according to the hyperparameter settings.

| Label smoothing | $\xi_{\text{TER}}$ | beam-width | $\mathcal{N}$ | BLEU4 avg | std |
|---|---|---|---|---|---|
| 0.1 | $\frac{\lambda}{\lambda_{\max}}$ | 1 | $\mathcal{U}(2,6)$ | 35.51 | 0.07 |
| 0.1 | $\frac{\lambda}{\lambda_{\max}}$ | 6 | $\mathcal{U}(2,6)$ | 35.68 | 0.09 |
| 0.1 | $\frac{\lambda}{\lambda_{\max}}$ | 3 | $\mathcal{U}(2,6)$ | **35.73** | 0.07 |
| 0.1 | $0.5\frac{\lambda}{\lambda_{\max}}$ | 3 | $\mathcal{U}(2,6)$ | 35.50 | 0.08 |
| 0.1 | $0.8\frac{\lambda}{\lambda_{\max}}$ | 3 | $\mathcal{U}(2,6)$ | 35.62 | 0.14 |
| 0.1 | $\frac{\lambda}{\lambda_{\max}}$ | 3 | $\mathcal{U}(2,4)$ | 35.67 | 0.12 |
| 0.1 | $\frac{\lambda}{\lambda_{\max}}$ | 3 | $\mathcal{U}(4,8)$ | 35.54 | 0.08 |
| 0.1 | 0.50 | 3 | $\mathcal{U}(2,6)$ | 35.47 | 0.10 |
| 0.1 | 0.99 | 3 | $\mathcal{U}(2,6)$ | 35.31 | 0.06 |
| 0.1 | $\infty$ | 3 | $\mathcal{U}(2,6)$ | 33.36 | 0.16 |
| 0.0 | $\frac{\lambda}{\lambda_{\max}}$ | 3 | $\mathcal{U}(2,6)$ | 35.19 | 0.11 |
| 0.1 | - | - | - | 34.65 | 0.12 |
| 0.0 | - | - | - | 33.94 | 0.17 |

