# OpenReview forum: "Minimum Edit Distance Training for Conditional Language Generation Models"
_ICLR.cc/2024/Conference — ICLR 2024 Conference Withdrawn Submission_

### Official Review · Reviewer_DVwK · 2023-10-29

**Soundness:** 2 fair
**Presentation:** 2 fair
**Contribution:** 1 poor
**Rating:** 3
**Confidence:** 4

**Summary:**

The paper proposes minimum edit distance training for machine translation (MT) and speech recognition (ASR) in the context of attention-based encoder-decoder models. The authors suggest that the conventional maximum likelihood training with teacher forcing results in exposure bias for these models, and as such, sequence-level objectives like minimum edit distance should work better since they are closely related to the evaluation metric. They also use the proposed objective to train a post-hoc calibration function. Experiments are mostly conducted using MT datasets, and a few using LibriSpeech (100h) for ASR, demonstrating improvements over the conventional training methods.

**Strengths:**

1. While minimum Bayes risk (MBR) training is a well-known method for ASR, it has not received adequate attention for MT, perhaps because it is hard to agree upon a good definition of “risk” for this task which is efficient to compute. As such, this paper should fill this gap. Since MT can have multiple correct “alignments”, the authors define a segment-level Levenshtein distance function (sMED) for this task.

2. Section 2 is effective (at the cost of being a little too verbose) in explaining the discrepancy between training and evaluation metrics, in terms of Hamming distance and Levenshtein distance.

3. The proposed MED training gives small BLEU score improvements on IWSLT data, and significant WER improvements on LibriSpeech (100h).

**Weaknesses:**

The two main weaknesses in the paper are novelty and empirical results, which I will describe in detail below.

### MBR training is well known

Minimum edit distance training is essentially a form of minimum Bayes risk (MBR) training where the risk function is Levenshtein distance between the reference and predicted sequences. MBR training has a long history in ASR, with instantiations in minimum phone error (MPE) [1], state-level minimum bayes risk (sMBR) [2], and MED training. In fact, the authors have cited [2] in the “Related Works” section (which is unfortunately delegated to the Appendix), but they fail to make the connection. Instead, they have strangely focused more on “scheduled sampling,” which is just a training hack to reduce overconfidence in models trained with teacher forcing. Aside from choices of the risk function, MBR training has been explored in the context of hybrid HMM-based models [2], RNN-transducers [3], and attention-based encoder-decoders [4]. The last one, in particular, uses MED training for the same type of models as proposed in the paper. The main difference between the strategy proposed in the paper versus these in literature is that the authors compute the risk against a single sequence obtained using autoregressive decoding. I would argue that this is, in fact, a weaker version of traditional MBR training, which computes the “expected” risk.

Even in machine translation, MBR has a rich history. It has been primarily used for decoding [5] but has also seen use in training of statistical MT models [6]. The authors have not situated their paper against any of these well-known works, which comes across as ignorance of prior work.

Once we take into account all these existing (but not cited) work, the remaining contribution of the paper is two-fold: (i) segment-level edit distance for MT, and (ii) using MED training for the calibration function. In my opinion, while (i) is a clever idea, it stands against the original motivation of reducing discrepancy between training and evaluation metrics — for example, [6] performed MBR training using BLEU score and showed that it improved BLEU metrics most, compared to edit distance based metrics. This only leaves (ii) — it is interesting to see improvement in calibration errors, but this is not sufficient by itself considering lack of novelty in the main training method.

[1] Povey, Daniel and Philip C. Woodland. “Minimum Phone Error and I-smoothing for improved discriminative training.” 2002 IEEE International Conference on Acoustics, Speech, and Signal Processing 1 (2002): I-105-I-108.

[2] Veselý, Karel et al. “Sequence-discriminative training of deep neural networks.” Interspeech (2013).

[3] Weng, Chao et al. “Minimum Bayes Risk Training of RNN-Transducer for End-to-End Speech Recognition.” Interspeech (2019).

[4] Cui, Jia et al. “Improving Attention-Based End-to-End ASR Systems with Sequence-Based Loss Functions.” 2018 IEEE Spoken Language Technology Workshop (SLT) (2018): 353-360.

[5] Kumar, Shankar and William J. Byrne. “Minimum Bayes-Risk Decoding for Statistical Machine Translation.” North American Chapter of the Association for Computational Linguistics (2004).

[6] Och, Franz Josef. “Minimum Error Rate Training in Statistical Machine Translation.” Annual Meeting of the Association for Computational Linguistics (2003).

### MT results show only minor BLEU improvements

The experimental results section is rather sparse, and the provided MT results do not suggest significant improvements. From Table 1, on most test sets, the BLEU score only improves by 0.5-1.0, with the relative improvement only about 2% in most cases. Furthermore, the relatively simple “scheduled sampling” already recovers 50% of this improvement in most cases, so I don’t see why practitioners would implement MED training at all.

### ASR experiments are done on a “toy” task

ASR results are only shown on the 100h subset of LibriSpeech, which is almost a toy task at this point. The authors should perform training and evaluation at least on the full 960h LibriSpeech. In Appendix A, they have mentioned that they used 4 A100 GPUs for the MT experiments — this hardware should be quite sufficient to conduct experiments also on the full LibriSpeech. Otherwise, readers may wonder whether these results were not shown because MED provided no improvement in this setting (which may very well be the case from my experience).

Besides the above limitations, I think it may be useful to reduce the space given to Section 2 (Background), since the train-eval discrepancy is already well known, and instead conduct more thorough evaluation.

**Questions:**

In Section 3.1, the LBP is essentially the Levenshtein alignment, which means that equation (11) would compute the edit distance (or WER) between the reference and decoded hypothesis. Is this correct?

---

### Official Review · Reviewer_kjTK · 2023-10-31

**Soundness:** 2 fair
**Presentation:** 1 poor
**Contribution:** 2 fair
**Rating:** 3
**Confidence:** 5

**Summary:**

Context: Attention-based encoder-decoder (AED) models (also referred to as conditional language generation (CLG) models) for speech recognition and machine translation.

A new training method is proposed to fix the exposure bias problem, in which the model only sees the reference label sequence as prefix during conventional training, maximum likelihood estimation (MLE).

The motivation is to avoid the exposure bias problem and to have the training loss more consistent with the evaluation metric, namely the word error rate, i.e. the edit distance / Levenshtein distance.

During training, beam search is performed to get a recognized sequence out. Then, the Levenshtein distance and alignment is calculated between the reference label sequence and the recognized label sequence. This gives a target for each position of the recognized sequence prefix, and that is used as an additional loss to the standard MLE training with teacher-forcing.

**Strengths:**

Interesting approach which should help for regularization and better generalization.

**Weaknesses:**

A lot of mathematical formulations are introduced, which are rather confusing and distracting from the method, and also have some problems (see details for an example). The method could be explained in a much simpler way.

The Optimal Completion Distillation (OCD) conceptually leads to a very similar loss. The Optimal Completion for any given prefix should be exactly what any possible Levenshtein alignment would give as the next possible targets for a given prefix. The Optimal Completion even solves the problem of ambiguity when there are multiple targets possible, by mapping that to a target probability distribution instead of having a single target. Given that this method is so similar, even looks almost the same (not taking the derivation/formulation into account, but just what you actually do in the end), this should be directly compared to, when presenting the method, to see the actual differences, and also in the experiments. This is missing here.

The novelty is also limited, given that OCD is so similar.

The ASR experiments are only done on the Librispeech 100h subset, which is way too small to be relevant. The whole Librispeech 960h should be used.

MBR training and OCD should be done for comparisons. I.e. not just citing results from another paper, but this should be done on the exact same model, within the same software, such that we really have a direct comparison.

No code is published?

ASR baseline is not good. (I did not really check the translation baseline too much. Maybe also not good.)
Many details missing.

**Questions:**

Definition 3.1: Something is wrong. You use union there, which is an operation for sets, but you use union on sequences, namely you take the union of ψ with the tuple (t+1,tˆ+1). I don't know what this is supposed to mean. Is this concatenation of sequences? Or is it union of sets?

Definition 3.3 and Section 3.3. I did not really understand this. I understand the problem, and I was also wondering how you choose the Levenshtein alignment actually when it is ambiguous, which it often is. I guess this is some sort of smoothing, but the way it is defined is too complicated.



> Because the inferred label sequence Yˆ i is used to calculate the LBP, the CLG model trained using 1:tˆ only NLL_{MED} in an offline scenario has a trivial solution that outputs the input label of the current time step.

I don't fully understand this. Why would it learn to just output the input label? That would lead to an insertion error.


> We trained the NMT model with the online scenario shown in Figure 2, and the ASR model with the offline scenario.

Why?

And what is really the difference in the definition mathematically? Is this just an implementation detail, or is there a real difference?

> In these experiments, we sampled the prefix from an interpolated distribution εSS P (Yˆ ) + (1 − εSS )P (Y )

I don't understand this. Is this on sequence level (Y being the whole sequence), or on label level (Y being one label)? What are the two distributions here? P is just one model, i.e. one distribution? Or it's on label level but using different prefixes? But this also does not make sense.

What models are used exactly? This seems to use existing recipes from Fairseq and ESPnet. So it would be helpful to reference them exactly? Or if these are not existing recipes, why not? And why are those specific models chosen? E.g. for the ASR model, you would rather choose a Conformer-based baseline, not a Transformer.


Runtime? Beam search during training might be expensive? So how much slower is the proposed training? This seems like a very relevant aspect, but there are no numbers on that?



The CY LIN. ROUGE... reference capitalization is wrong, it's Chin-Yew Lin. 2004. ROUGE...

---

### Official Review · Reviewer_FobE · 2023-11-02

**Soundness:** 3 good
**Presentation:** 3 good
**Contribution:** 3 good
**Rating:** 6
**Confidence:** 4

**Summary:**

This paper introduces a modification to the conventional Maximum Likelihood Estimate (MLE) training method, involving teacher forcing for encoder-decoder architectures, with a focus on minimum edit distance-based training. The paper's argument centers on the distinction between MLE training and inference, emphasizing that, during MLE training with teacher forcing, only corrections and substitutions are addressed, while deletion and insertion errors are not considered. In contrast, evaluation metrics, such as edit distances (WER) or n-gram overlaps (Bleu), differ from the training-time evaluation.

The core concept of this paper involves the training process, where a target sequence is initially sampled during training, using a beam search either from the same model or a different one. Subsequently, a Levenshtein alignment path is computed between these sequences. The MLE loss is then calculated over the Levenshtein backward path, accounting for insertion and deletion errors. For Neural Machine Translation (NMT) tasks, the paper introduces a segment-level modification, where the original sequence is segmented into n-grams, and alignment is established accordingly. The paper contends that Minimum Edit Distance (MED) training also enhances the model's calibration.

Empirical results from experiments conducted on NMT and ASR tasks underscore the practical benefits of the proposed minimum edit distance-based training approach.

**Strengths:**

1. The paper is well-written and easy to follow.

2. The empirical results show the usefulness of the approach.

3. The calibration results are useful.

**Weaknesses:**

1. Lack of comparison with minimum edit distance based training in ASR systems. The idea of using the edit distance criteria for training the ASR model is quite well explored [1,2].




References:

[1] Prabhavalkar, Rohit, et al. "Minimum word error rate training for attention-based sequence-to-sequence models." 2018 IEEE International Conference on Acoustics, Speech and Signal Processing (ICASSP). IEEE, 2018.

[2] Tian, Jinchuan, et al. "Integrating lattice-free MMI into end-to-end speech recognition." IEEE/ACM Transactions on Audio, Speech, and Language Processing 31 (2022): 25-38.

**Questions:**

1. How does sMED compare against MED for the NMT tasks?

2. How much is the training speed impacted by MED computations? For MBR training, it is one of the major problem?

3. What are the results of online training with MED for ASR task?

4. Can you share examples where sMED objective lead to superior target sequences?

---

### Official Review · Reviewer_eZjX · 2023-11-10

**Soundness:** 3 good
**Presentation:** 3 good
**Contribution:** 3 good
**Rating:** 6
**Confidence:** 4

**Summary:**

The paper proposed an approach to train AED models to minimize the edit distance between the predicted sequence and the reference sequence. It changes the teacher forcing training to condition on the beam search decoding history instead of ground-truth history, and predict the next correct token along the alignment path with the minimum edit distance between the beam search decoded sequence and the reference sequence. By doing this, it can mitigate the expose bias and optimize the evaluation metric during training, leading to a better generalization performance. The proposed method is experimented in NMT and ASR tasks, showing improved performance.

**Strengths:**

- The proposed idea is a new approach to mitigate the exposure bias issue, and optimize sequence level evaluation metric.
- The proposed method is intuitive and technically sound.
- It is a relatively straight-forward change in the training implementation.
- Consistent improvement over the MLE approach across all NMT and ASR tasks in the experiments.
- Calibration function performance also benefits from the proposed minimum edit distance loss, which in turn helps reduce WER as well.
- The analysis showing the proposed approach reduces the accumulated errors due to exposure bias, especially for longer sequences, is promising.
- Presentation is clear.

**Weaknesses:**

- There are already many methods to optimize the evaluation metrics in the sequence-level training, as the authors mention some of them. E.g. there are minimum WER and MBR based approaches for ASR, see some examples below. But they are not compared in the ASR experiments in the paper. The proposed method was mainly compared to MLE, which is a weaker baseline. Even for NMT, only one task shows MBR result. This seems to be a major drawback of the paper.

Prabhavalkar et al., Minimum Word Error Rate Training for Attention-based Sequence-to-Sequence Models, 2018
Shannon, Optimizing expected word error rate via sampling for speech recognition, 2017
Kingsbury, Lattice-based optimization of sequence classification criteria for neural-network acoustic modeling, 2009

- Table 2 shows the results for different calibration functions. It seems they are all done in the ASR model that was trained with MLE. Can the same comparisons be done in the ASR model trained with MED as well? Would the MED trained calibration still improve the MED trained ASR model performance further?

**Questions:**

- The paper mentions "Because the inferred label sequence is used to calculate the LBP, the CLG model trained using
only NLL_MED in an offline scenario has a trivial solution that outputs the input label of the current time step." But according to Eq (11), by definition NLL_MED trains the model to predict the correct token at the next time step in the LBP based on the previous predicted sequence. How come the model will trivially learn to output the input label of the current time step? Please elaborate.

- Some brackets in Eq (13) seem to be missing. The notation is not clear enough.